# Adaptive Packet Coding for Reliable Underwater Acoustic Communications

**Lianyou Jing** [1,2,*] 🔾, **Yongqi Tang** [1], **Chengbing He** [3] **and Hongxi Yin** [1,2]

1 School of Information and Communication Engineering, Dalian University of Technology, Dalian 116024, China
2 The State Key Laboratory of Integrated Services Networks, Xidian University, Xi'an 710071, China
3 School of Marine Science and Technology, Northwestern Polytechnical University, Xi'an 710072, China
* Correspondence: lyjing@dlut.edu.cn

**Abstract:** This work investigates adaptive random linear packet coding (RLPC) for reliable underwater acoustic (UWA) communications. Our goal is to minimize the total transmission time of data blocks by adjusting the packet coding rate. We first consider the application of RLPC with the conventional automatic repeat request (ARQ) scheme. We dynamically adjust the coding rate to fit the time variations of UWA channels by choosing the optimal number of packets in each transmission. The optimal number of packets in each transmission is obtained based on a dynamic programming (DP) algorithm according to the feedback messages, which contain the number of successfully transmitted packets in the last transmission and the channel state information. Furthermore, considering the long propagation delay of UWA communications, we propose a modified juggling-like ARQ (J-ARQ) for the RLPC scheme, for which the duration of each transmission can be adjusted based on the characteristics of RLPC. A two-step DP algorithm is proposed to find out the optimal solutions for this case. Simulation results show that the proposed schemes can improve the throughput efficiency and reduce the outage probability.

**Keywords:** underwater acoustic communications; random linear packet coding; dynamic programming; juggling-like ARQ

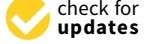



## 1. Introduction

Underwater communications have attracted much interest in recent years due to the rapid development of ocean exploration. Reliable data transmission among nodes is the basis of many underwater applications, such as tsunami warning, ecological monitoring, off-shore oil exploration, etc. [1]. Compared to other media, acoustic waves are the only means of achieving reliable underwater transmission over a long distance. However, underwater acoustic (UWA) communications also face some difficulties compared to their terrestrial counterparts. The characteristics of UWA channels include large delay and Doppler spread, high path attenuation, limited bandwidth, and long propagation delay, which severely reduce the reliability of underwater transmissions.

Generally speaking, redundancy and retransmission are the two main approaches to enable a robust communication link. Retransmission schemes such as automatic repeat request (ARQ) are widely used in terrestrial wireless communications. However, the long propagation delay caused by the slow speed of acoustic waves (approximately 1500 m/s) in water makes the standard ARQ scheme very inefficient in UWA communications. Thus, there have been many modified ARQ schemes proposed for UWA communications, such as the go-back $N$ scheme, the stop-and-wait (S&W) scheme and the selective repeat (SR) scheme [2,3]. These modified ARQ schemes improve the efficiency compared to standard ARQ, but not well enough. This is because the sender always remains idle when it is waiting for the receiver's acknowledgement (ACK) messages in these schemes. As a result,

they are still quite inefficient when the propagation time is long enough. In [4,5], a juggling-like ARQ scheme (J-ARQ) is proposed to further improve the channel utilization, where the sender reserves a fixed gap for the ACK reception after each transmission. In this way, the sender still could transmit the data when it is awaiting the ACK. However, the fixed transmission and reception duration are used in the J-ARQ scheme to avoid conflict, which will lead to a decrease in transmission efficiency when the transmitter does not have an infinite amount of data waiting to be transmitted [5].

Compared to retransmission, the redundancy strategy alternatively seeks an attempt to improve the quality of each transmission. A common redundancy technique is forward error control (FEC), which can lower the bit error rate (BER) at the cost of reducing the data rate. For example, bit-level channel coding schemes, such as low-density parity codes [6] and convolutional codes [7], have been widely used in the physical layer of UWA communications. In contrast, in this paper we investigate another FEC technique, which is performed on the packet level and designed for the link layer [8–17]. Its mechanism is that the sender sends multiple coded packets which are encoded by original packets. The receiver can recover the original packets correctly if there are enough coded packets successfully received. Rateless codes [18–20], as a good packet-coding technology, have been used in UWA communications. These codes are so named since the sender can generate unlimited coded packets to ensure that the original packets can be fully recovered. In [12], a kind of rateless code, Raptor code, is used in UWA communications. The coding rate is optimized to maximize the throughput over UWA channels. In [13], a cross-layer FEC scheme which combines both the physical layer FEC and the packet layer FER is extended based on [12].

So far, there have been many works which combine the redundancy strategy and retransmission strategy based on the rateless codes. In [5], rateless code is used in the J-ARQ scheme. In [10], an HARQ scheme is proposed, named segmented data reliable transport (SDRT), to achieve reliable data transmission in UWA sensor networks by employing Tornado codes. In this scheme, the sender keeps sending the packets until it receives an ACK. In order to reduce the energy consumption for unnecessary transmission, a window control mechanism is further proposed to estimate the expected number of packets actually needed. With the information, the sender just transmits a pre-estimated number of encoder packets (i.e., the window size), and then slows down the transmission to wait for an ACK. In [11], an underwater hybrid ARQ (UW-HARQ) is proposed which uses an NACK to feed back the receive state. Random linear packet coding (RLPC) is used as a rateless code in [14–16]. In [14], the optimal number of coded packets is investigated for a half-duplex link to minimize the time (or energy) required for the transmission of a group of packets. In [15], joint power and rate control for an acoustic link employing random linear packet coding is considered to achieve a prespecified outage/reliability criterion. To this end, ref. [16] extends the work in [15] by grouped packet coding and increases its throughput efficiency based on the S&W ARQ scheme. The sender transmits a super-group packets and waits for an ACK. This could improve efficiency by packing multiple packets together.

In this paper, we propose two adaptive packet-coding schemes to achieve reliable UWA communications. We consider the time variations of UWA channels, which are ignored in most works. Since we focus on the link level performance, we model the channel state by a finite-state Markov chain (FSMC) [21]. Some works use FSMC to model the UWA channels [22–28]. In [22,23], Preisig used a two-state FSMC channel model with known transition probabilities to evaluate energy-efficient schedulers for underwater acoustic point-to-point links based on experimental data. As compared to existing works, our main contributors are summarized as follows:

(1) We first propose an S&W ARQ scheme with RLPC (RLPC-ARQ). In the RLPC-ARQ scheme, the number of successfully transmitted packets in each transmission and the channel state information (CSI) are fed back to the sender in the ACK messages. The sender dynamically adjusts the coding rate based on the ACK messages to fit the time variations of UWA channels by choosing the optimal number of packets

in each transmission. Meanwhile, considering the long propagation delay and the error-prone characteristic of UWA communications, we set a maximum number of retransmissions to avoid infinite retransmission. The problem is formulated as a finite horizon optimization problem. A dynamic programming (DP) algorithm is proposed to obtain the optimal number of packets in each transmission.

(2) We also propose a modified juggling-like ARQ scheme for the RLPC system (RLPC-J-ARQ) to deal with the long propagation of UWA channels. In the RLPC-J-ARQ scheme, two data blocks are alternately transmitted at the sender; there is no need to stop and wait for the ACK. Different from the J-ARQ scheme in [5] for which the duration of each transmission is fixed, the RLPC-J-ARQ scheme adopts adjustable transmission duration by exploiting the characteristics of the rateless code. The rateless code is only used when there is not enough data to transmit to avoid idleness. In addition, we also consider the effect of channel variation. In this case, the standard DP algorithm does not work. Thus, we also propose a two-step DP algorithm to find out the optimal number of packets in each transmission.

The rest of this paper is organized as follows. Section 2 presents the basic system model. We formulate the optimization problem in Section 3. We propose the optimization solution based on the principle of DP in Section 4. Section 5 proposes the RLPC-J-ARQ scheme. Section 6 proposes a two-step DP approach to find out the optimal solution of the RLPC-J-ARQ scheme. Section 7 provides numerical results for the proposed solutions and Section 8 concludes the paper.

## 2. System Model

We consider a point-to-point system operating in a half-duplex manner without interference from other nodes. The sender collects $W_s$ information packets. Each packet has $N_b$ symbols and $N_b$ is a constant number during the transmission. These information packets are first divided into multiple blocks, each containing $W$ packets. In this paper, we only consider the FEC technique in the packet level. The $W$ original packets in each block are then encoded based on the RLPC [14–16] to generate sufficient coded packets. We assume that the data is delay constrained, which means the receiver should receive the data before the deadline. Otherwise, the data becomes useless.

We first consider an S&W ARQ scheme with RLPC, which is illustrated in Figure 1. The coded packets from different blocks are transmitted in sequence. Let $N_i^{(k)}$ denotes the number of transmitted packets for the $k$-th block in the $i$-th transmission. Let $\rho_{k,i}^{(1)}$ and $\rho_{k,i}^{(2)}$ denote the beginning and ending time instants of the $k$-th block in the $i$-th transmission. Hence, the relationship between them is

$$\rho_{k,i}^{(2)} = \rho_{k,i}^{(1)} + T_p N_i^{(k)} \tag{1}$$

where $T_p = N_b T_s$ is the time duration of a coded packet, $T_s$ is the symbol duration.

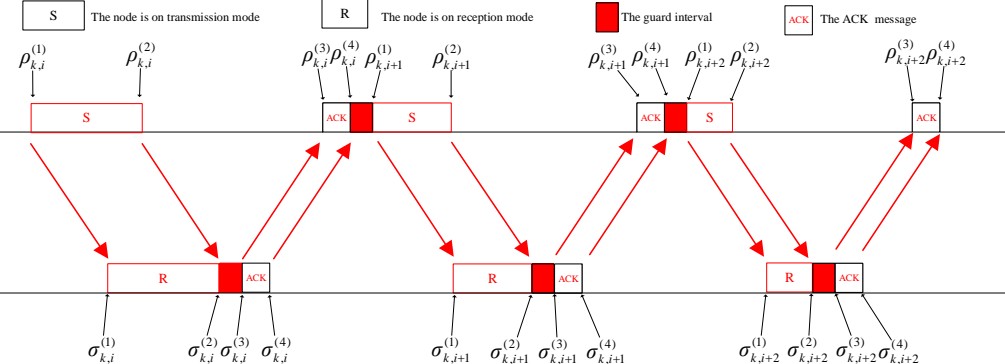

**Figure 1.** The S&W ARQ scheme with RLPC.

Let $T_d$ denote the propagation delay. Then, the transmitted signal is received after $T_d$ duration, $T_d = d/c$, where $d$ is the distance between the transceiver pair, and $c$ is the sound speed in water. Let $\sigma_{k,i}^{(1)}$ and $\sigma_{k,i}^{(2)}$ denote the beginning and ending time instants of the received signal at the receiver. Then, we have

$$\begin{cases} \sigma_{k,i}^{(1)} = \rho_{k,i}^{(1)} + T_d \\ \sigma_{k,i}^{(2)} = \sigma_{k,i}^{(1)} + T_p N_i^{(k)} \end{cases} \tag{2}$$

When the receiver completes the receiving process, there is a guard interval, $T_g$, for the receiver node to process the data and change the mode. Next, the receiver responds with an ACK. According to the characteristic of RLPC, the receiver can recover the original information if there are enough coded packets successfully received. Thus, we do not need to retransmit the unsuccessful packets in the following transmission. In the following transmission, the sender will send enough other coded packets. It is noted that the other coded packets are also generated from the same information packets. There is the main different from the traditional ARQ scheme. Hence, the ACK contains the number of successfully received packets $s_i^{(k)}$ in the current transmission and some pilot symbols used for channel estimation at the sender.

The BER $e$ depends on the channel coding rate, the modulation order, the channel state, and the codeword length. In this paper, these parameters are fixed except the channel state. Thus, the packet error probability (PEP) $P_c$ is only connected with the channel state.

As mentioned above, we focus on the link level performance, so we model the channel state by a finite-state Markov chain (FSMC) [22,24,25]. Assume the channel state is constant during the $i$-th transmission. Let $h_i^{(k)}$ denote the CSI when the $k$-th block is transmitted at the $i$-th transmission, which is quantized into $L$ levels, i.e., $h_i^{(k)} \in \{\delta_1, \ldots, \delta_L\}$. The state transition probabilities $P_h(h_i^{(k)}|h_{i-1}^{(k)})$ are assumed known.

Let $P_s(s_i^{(k)}, h_i^{(k)}, N_i^{(k)})$ denote the probability that $s_i^{(k)}$ coded packets have been successfully received when $N_i^{(k)}$ coded packets are transmitted under the channel state $h_i^{(k)}$. Then,

$$P_s(s_i^{(k)}, h_i^{(k)}, N_i^{(k)}) = \binom{s_i^{(k)}}{N_i^{(k)}} (1 - P_c)^{s_i^{(k)}} P_c^{(N_i^{(k)} - s_i^{(k)})} \tag{3}$$

Let $\sigma_{k,i}^{(3)}$ and $\sigma_{k,i}^{(4)}$ denote the beginning and ending time points for the ACK at the receiver. Then, the relationship is given as

$$\begin{cases} \sigma_{k,i}^{(3)} = \sigma_{k,i}^{(2)} + T_g \\ \sigma_{k,i}^{(4)} = \sigma_{k,i}^{(3)} + T_{\text{ACK}} \end{cases} \tag{4}$$

where $T_{\text{ACK}}$ is the time duration of ACK.

The sender will receive the corresponding ACK after $T_d$ time duration. Let $\rho_{k,i}^{(3)}$ and $\rho_{k,i}^{(4)}$ denote the beginning and ending time points for the received ACK at the sender. Then,

$$\begin{cases} \rho_{k,i}^{(3)} = \sigma_{k,i}^{(3)} + T_d \\ \rho_{k,i}^{(4)} = \rho_{k,i}^{(3)} + T_{\text{ACK}} \end{cases} \tag{5}$$

The sender could estimate the CSI based on the pilot symbols in the ACK and decode the feedback messages $s_i^{(k)}$. We use a low rate channel code for ACK messages to improve the reliability of feedback messages. To reduce the complexity of the problem, we assume the feedback messages are error-free.

There is also a guard interval for the sender to process the ACK and change the mode. In this way, the $i$-th transmission for the $k$-th block is done. Hence, the total time duration for the $i$-th transmission is given as

$$T_i^{(k)} = T_p N_i^k + 2T_d + 2T_g + T_{\text{ACK}} \tag{6}$$

$$= T_p N_i^k + T_\tau. \tag{7}$$

where $T_\tau = 2T_d + 2T_g + T_{\text{ACK}}$ is fixed for each transmission.

If the receiver has received enough coded packets to recover the original information from the $k$-th block, the sender will send the coded packets from another block in the next transmission. Otherwise, the sender determines the number of transmitted coded packets in the $(i+1)$-th transmission based on the feedback ACK. Thus, the data rate of packet coding is not constant during the transmission. The sender dynamically adjusts the rate based on the current transmission result to adapt to the time variation of UWA channels.

## 3. Problem Formulation

Assume that the receiver has successfully demodulated $q_{i-1}^{(k)}$ packets from the $k$-th block in the previous $i-1$ transmissions according to the feedback messages. Then, $q_i^{(k)}$ is given by

$$q_i^{(k)} = \sum_{j=1}^{i} s_j^{(k)} = q_{i-1}^{(k)} + s_i^{(k)} \tag{8}$$

According to the characteristics of RLPC, the information packets can be recovered correctly if [15,16]

$$q_i^{(k)} \geq W \tag{9}$$

Then, according to (8), we have

$$s_i^{(k)} \geq W - q_{i-1}^{(k)} \tag{10}$$

Let $u_i^{(k)} = W - q_{i-1}^{(k)}$ denote the minimum number of packets needed to guarantee correct decoding. Thus, the probability of $W$ information packets can be received correctly after the $i$-th transmission is given by

$$P_S^{(i)} = \sum_{s_i^{(k)}=u_i^{(k)}}^{N_i^{(k)}} P_s(s_i^{(k)}, h_i^{(k)}, N_i^{(k)}) \tag{11}$$

It is obvious that the number of transmitted packets during the $i$-th transmission, $N_i^{(k)}$, cannot exceed a maximal level, denoted by $B_i^{(k)}$. Then,

$$0 \leq N_i^{(k)} \leq B_i^{(k)} \tag{12}$$

Our objective is to minimize the total transmission time of data blocks for delay-constrained applications by adjusting the packet-coding rate. Since the packet error is a stochastic process, there are certainly cases where the information cannot be received correctly after many transmissions, especially for error-prone UWA channels. However, considering the significant propagation delay of UWA channels, too many retransmissions is unacceptable in a practical communications system. Thus, we should set a maximal number of retransmissions, $M$.

To make sure that $W$ information packets can be received correctly after $M$ transmissions, we add a penalty term for the $M$-th transmission. In this way, the transmission time of the $M$-th transmission is given by

$$T_M^{(k)} = T_p N_M^{(k)} + T_\tau + \pi(q_M^{(k)}) \tag{13}$$

where

$$\pi(q_M^{(k)}) = \begin{cases} 0, & q_M^{(k)} \geq W \\ C, & q_M^{(k)} < W \end{cases} \tag{14}$$

$C$ is a large value. This setting implies that if the information cannot be recovered correctly after $M$ transmissions, it will incur a large cost.

Then, the objective function is given by

$$J_T = \min_{N_i^{(k)}} \sum_{i=1}^{M} \gamma^i T_i^{(k)} \tag{15}$$

where $\gamma$ is discount factor. Consider the cost of storage space and number overhead, $\gamma > 1$.

Naturally, there is a trade-off between the coding rate and the transmission time. Generally speaking, sending more coded packets will improve the reliability of each transmission. Moreover, it will reduce the probability of retransmission and then reduce the total transmission time. However, if we send too many packets in each transmission, the sending time also increases, which also decreases efficiency. Thus, there is an optimal coding rate for each transmission to minimize the total transmission time.

## 4. Optimal Solution

In our problem, the state of the $i$-th transmission is dependent on the result of $(i-1)$-th transmission. Meanwhile, since packet error is a stochastic process, this problem can be seen as a sequential decision-making problem. This problem can be solved by the finite-horizon DP [24,25,29] approach.

Let $c_i^{(k)} = (h_{i-1}^{(k)}, q_{i-1}^{(k)})$ denote the state of the system at the $i$-th transmission for the $k$-th block, where $q_{i-1}^{(k)}$ denotes the number of coded packets successfully received in the all $(i-1)$ transmissions. $h_{i-1}^{(k)}$ is the feedback channel state of the $(i-1)$-th transmission for the $k$-th block. Let $U(c_i^{(k)})$ denote the feasible set of the *action* $N_i^{(k)}$ given the current state $c_i^{(k)}$. Then,

$$U(c_i^{(k)}) = [0, B_i^{(k)}] \tag{16}$$

In this case, the upper limit of $N_i^{(k)}$ is constant. Thus, we let $B_i^{(k)} = Q$.

The probability that the system state $c_i^{(k)}$ will transfer to state $c_{i+1}^{(k)}$ when action $N_i^{(k)}$ is taken is given as

$$\begin{aligned} P(c_{i+1}^{(k)}|c_i^{(k)}, N_i^{(k)}) &= P(h_i^{(k)}, q_i^{(k)}|h_{i-1}^{(k)}, q_{i-1}^{(k)}, N_i^{(k)}) \\ &= P(q_i^{(k)}|h_i^{(k)}, q_{i-1}^{(k)}, N_i^{(k)}) P(h_i^{(k)}|h_{i-1}^{(k)}) \end{aligned} \tag{17}$$

According to (8),

$$P(q_i^{(k)}|h_i^{(k)}, q_{i-1}^{(k)}, N_i^{(k)}) = \begin{cases} P_s(s_i^{(k)}, h_i^{(k)}, N_i^{(k)}), & q_i^{(k)} \geq q_{i-1}^{(k)} \\ 0, & q_i^{(k)} < q_{i-1}^{(k)} \end{cases} \tag{18}$$

The expected cost incurred from the $M$-th transmission is given by

$$
\begin{aligned}
&J_M(h_{M-1}^{(k)}, q_{M-1}^{(k)}) \\
&= \min_{0 \le N_M^{(k)} \le Q} \left\{ \gamma^M T_M^{(k)} + \mathbb{E}\left[ \pi(q_M^{(k)}) | (h_{M-1}^{(k)}, q_{M-1}^{(k)}) \right] \right\}
\end{aligned}
\tag{19}
$$

$$
= \min_{0 \le N_M^{(k)} \le Q} \left\{ \gamma^M T_p N_M^{(k)} + \gamma^M T_\tau + C \sum_{s_M^{(k)}=0}^{u_M^{(k)}} \sum_{l=1}^{L} P_s(s_M^{(k)}, h_M^{(k)}, N_M^{(k)}) P(h_M^{(k)} = \delta_l | h_{M-1}^{(k)}) \right\}
\tag{20}
$$

The first term of (19) is the cost of the $M$-th transmission, the second term is the expectation of penalty term due to the incomplete transfer after $M$ transmissions. It depends on the result of the $M$-th transmission according to (14).

Based on the Bellman equation [29], the optimal solution can be obtained by recursively computing $J_M(c_M^{(k)})$, $J_{M-1}(c_{M-1}^{(k)})$, $\cdots$, $J_1(c_1^{(k)})$. The expected cost incurred from the $i$-th transmission to termination is given by

$$
\begin{aligned}
&J_i(h_{i-1}^{(k)}, q_{i-1}^{(k)}) \\
&= \min_{0 \le N_i^{(k)} \le Q} \left\{ \gamma^i T_i^{(k)} + \gamma \mathbb{E}\left[ J_{i+1}(h_i^{(k)}, i, q_i^{(k)}) | q_{i-1}^{(k)}, h_{i-1}^{(k)} \right] \right\} \\
&= \min_{0 \le N_i^{(k)} \le Q} \left\{ \gamma^i (T_p N_i^{(k)} + T_\tau) + \gamma \sum_{s_i^{(k)}=0}^{N_i^{(k)}} \sum_{l=1}^{L} P_s(s_i^{(k)}, h_i^{(k)}, N_i^{(k)}) P(h_i^{(k)} = \delta_l | h_{i-1}^{(k)}) J_{i+1}(h_i^{(k)}, q_i^{(k)}) \right\}
\end{aligned}
\tag{21}
$$

The first term in (21) represents the expected cost of current transmission. The second term in (21) is the expected future cost accumulated from the $(i+1)$-th transmission to the $M$-th transmission.

The DP approach includes two steps. In step 1, the sender calculates the (21) to find the optimal $N_i^{(k)*}$ for each system state $c_i^{(k)}$ and records it as a lookup table. In step 2, the sender chooses the optimal $N_i^{(k)*}$ of the current state based on the lookup table.

## 5. RLPC-J-ARQ Scheme

In the RLPC-ARQ scheme, the sender idles for a round-trip time while waiting for the ACK. This is quite inefficient for UWA communications, which have a long propagation delay. Ref. [5] proposed the J-ARQ to solve this problem. However, the block size of J-ARQ is fixed, which leads to the analysis in [5]; it cannot be directly applied in this problem. Essentially, the J-ARQ scheme uses the time that the feedback information is transmitted in the channel to send the following data blocks. According to this idea, we propose a RLPC-J-ARQ in our problem.

Figure 2 illustrates the transmission style of the RLPC-J-ARQ scheme. We only consider that two blocks are transmitted simultaneously. The sender leaves a gap after sending $N_i^{(k)}$ coded packets for receiving the ACK messages. Similarly, a guard interval is reserved for the node to change the mode. Next, the sender sends $N_i^{(k+1)}$ data packets for the $(k+1)$-th block. In this way, the transmitter could send more data blocks than the RLPC-ARQ scheme.

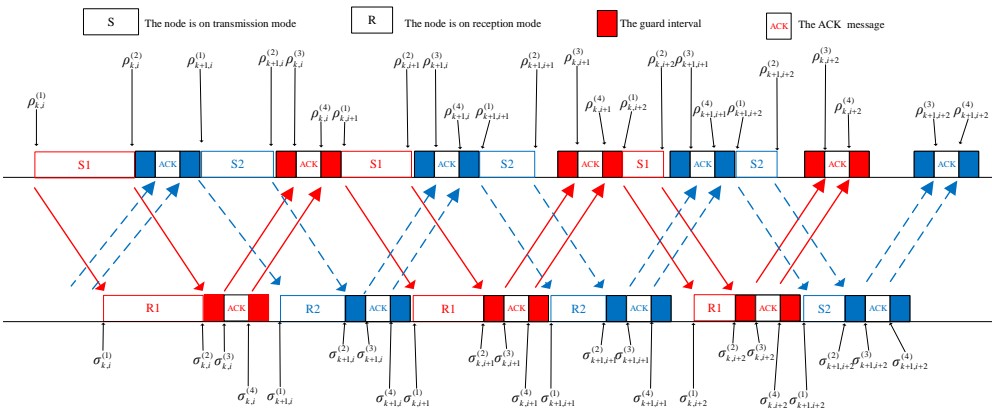

**Figure 2.** The proposed RLPC-J-ARQ scheme.

In Figure 2, $\rho_{k+1,i}^{(j)}$ and $\sigma_{k+1,i}^{(j)}$ represent the time points for the sender and receiver with the $(k+1)$-th block, respectively. The relationship between the time instants $\rho_{k+1,i}^{(j)}$ and $\sigma_{k+1,i}^{(j)}$ are similar to those of the $k$-th block. According to Figure 2, there are some constraints about the time instants to avoid the collision. At the sender, we have

$$\begin{cases} \rho_{k+1,i}^{(2)} \leq \rho_{k,i}^{(3)} - T_g \\ \rho_{k,i+1}^{(2)} \leq \rho_{k+1,i}^{(3)} - T_g \\ \rho_{k+1,i+1}^{(2)} \leq \rho_{k,i+1}^{(3)} - T_g \\ \rho_{k,i+2}^{(2)} \leq \rho_{k+1,i+1}^{(3)} - T_g \end{cases} \tag{22}$$

At the receiver, we have

$$\begin{cases} \sigma_{k,i}^{(4)} \leq \sigma_{k+1,i}^{(1)} - T_g \\ \sigma_{k+1,i}^{(4)} \leq \sigma_{k,i+1}^{(1)} - T_g \\ \sigma_{k,i+1}^{(4)} \leq \sigma_{k+1,i+1}^{(1)} - T_g \\ \sigma_{k+1,i+1}^{(4)} \leq \sigma_{k,i+2}^{(1)} - T_g \end{cases} \tag{23}$$

According to (1), (2), (4) and (5), the relationship between $\rho_{k,i}^{(j)}$ and $\rho_{k,i+1}^{(j)}$ is given as

$$\begin{cases} \rho_{k,i+1}^{(1)} = \rho_{k,i}^{(1)} + T_\tau + T_p N_i^{(k)} \\ \rho_{k,i+1}^{(2)} = \rho_{k,i}^{(2)} + T_\tau + T_p N_{i+1}^{(k)} \\ \rho_{k,i+1}^{(3)} = \rho_{k,i}^{(3)} + T_\tau + T_p N_{i+1}^{(k)} \\ \rho_{k,i+1}^{(4)} = \rho_{k,i}^{(4)} + T_\tau + T_p N_{i+1}^{(k)} \end{cases} \tag{24}$$

At the receiver, there is the same relationship $\sigma_{k,i}^{(j)}$ and $\sigma_{k,i+1}^{(j)}$. Similarly, there is the same relationship for the $(k+1)$-th block.

Thus, according to the relationship in (24), (22) can be rewritten as

$$\begin{cases} \rho_{k+1,i}^{(2)} \leq \rho_{k,i}^{(3)} - T_g \\ \rho_{k,i+1}^{(2)} \leq \rho_{k+1,i}^{(3)} - T_g \\ \rho_{k+1,i}^{(2)} + N_{i+1}^{(k+1)} \leq \rho_{k,i}^{(3)} - T_g + T_p N_{i+1}^{(k)} \\ \rho_{k,i+1}^{(2)} + N_{i+2}^{(k)} \leq \rho_{k+1,i}^{(3)} - T_g + T_p N_{i+1}^{(k+1)} \end{cases} \tag{25}$$

After further straightforward deduction, we have

$$
\begin{cases}
\rho_{k+1,i}^{(2)} \leq \rho_{k,i}^{(3)} - T_g \\
\rho_{k,i+1}^{(2)} \leq \rho_{k+1,i}^{(3)} - T_g \\
N_{i+1}^{(k+1)} \leq N_{i+1}^{(k)} \\
N_{i+2}^{(k)} \leq N_{i+1}^{(k+1)}
\end{cases}
\tag{26}
$$

Similar results can be obtained for the receiver, (23) can be rewritten as

$$
\begin{cases}
\sigma_{k,i}^{(4)} \leq \sigma_{k+1,i}^{(1)} - T_g \\
\sigma_{k+1,i}^{(4)} \leq \sigma_{k,i+1}^{(1)} - T_g \\
N_{i+1}^{(k)} \leq N_i^{(k+1)} \\
N_{i+1}^{(k+1)} \leq N_{i+1}^{(k)}
\end{cases}
\tag{27}
$$

For the first transmission, $i = 1$, the constraints at the sender are given by

$$
\begin{cases}
\rho_{k+1,1}^{(2)} \leq \rho_{k,1}^{(3)} - T_g \\
\rho_{k,1}^{(2)} \leq \rho_{k+1,1}^{(3)} - T_g \\
\sigma_{k,1}^{(4)} \leq \sigma_{k+1,1}^{(1)} - T_g \\
\sigma_{k+1,1}^{(4)} \leq \sigma_{k+1,2}^{(1)} - T_g
\end{cases}
\tag{28}
$$

Let $\rho_{k,1}^{(1)} = 0$; according to (28), the constraints about the initial time instant of the $(k+1)$-th block are given as

$$
\begin{cases}
\rho_{k+1,1}^{(1)} \leq 2T_d + T_p N_1^{(k)} - T_p N_1^{(k+1)} \\
\rho_{k+1,1}^{(1)} \geq T_p N_1^{(k)} + 2T_g + T_{\text{ACK}}
\end{cases}
\tag{29}
$$

We set $\rho_{k+1,1}^{(1)} = T_p N_1^{(k)} + 2T_g + T_{\text{ACK}}$. Namely, the sender sends the packets from the $(k+1)$-th block immediately once it is allowed to send. Meanwhile, the number of packets from the $(k+1)$-th block is also restricted,

$$
N_1^{(k+1)} \leq (2T_d - 2T_g - T_{\text{ACK}})/T_p
\tag{30}
$$

To sum up, the constraints in our problem are as follows

$$
\begin{cases}
N_1^{(k+1)} \leq (2T_d - 2T_g - T_{\text{ACK}})/T_p \\
N_{i+1}^{(k)} \leq N_i^{(k+1)} \\
N_{i+1}^{(k+1)} \leq N_{i+1}^{(k)} \\
1 \leq i \leq M
\end{cases}
\tag{31}
$$

According to (31), the number of transmitted packets for the $(k+1)$-th block at the $i$-th transmission does not exceed that of for the $k$-th block at the $i$-th transmission. The number of transmitted packets for the $k$-th block at the $(i+1)$-th transmission doe not exceed that of for the $(k+1)$-th block at the $i$-th transmission. If $N_i^{(k)} = 0$ or $N_i^{(k+1)} = 0$, of course, there are no constraints about the $N_i^{(k+1)}$ or $N_{i+1}^{(k)}$. Thus, the maximum number of transmitted packets in the $(i+1)$-th transmission are given by

$$B_{i+1}^{(k)} = \begin{cases} Q, & \text{if } N_i^{(k+1)} = 0 \\ N_i^{(k+1)}, & \text{if } N_i^{(k+1)} > 0 \end{cases} \tag{32}$$

$$B_{i+1}^{(k+1)} = \begin{cases} Q, & \text{if } N_{i+1}^{(k)} = 0 \\ N_{i+1}^{(k)}, & \text{if } N_{i+1}^{(k)} > 0 \end{cases} \tag{33}$$

There is a difference from the RLPC-ARQ scheme. In the RLPC-ARQ scheme, the sender can adjust the number of transmitted packets freely based on the feedback ACK. However, in this case, the number of transmitted packets in each transmission is monotone decreasing.

The transmission time duration for the $i$-th transmission is also different. For the first transmission, the transmission time duration is given by

$$T_1 = \rho_{k+1,2}^{(1)} = T_p N_{k,1} + 2T_g + T_{\text{ACK}} + T_p N_{k+1,1} + T_\tau \tag{34}$$

For the $i$-th ($1 < i < M$) transmission, if $N_i^{(k+1)} > 0$, according to Figure 2, the transmission time duration for the $i$-th transmission is given by

$$T_i = \rho_{k+1,i+1}^{(1)} - \rho_{k+1,i}^{(1)} = T_p N_{k+1,i} + T_\tau \tag{35}$$

In this case, it seems that the transmission time duration for the $i$-th transmission has nothing to do with the number of transmitted packets from the $k$-th block. However, $N_i^{(k)}$ still affects the transmission time duration according to the constraints in (31).

If $N_{k,i} > 0$ and $N_{k+1,i} = 0$, the transmission time duration should be

$$T_i = \rho_{k,i+1}^{(1)} - \rho_{k+1,i}^{(1)} \tag{36}$$

However, it is difficult to calculate $\rho_{k,i+1}^{(1)} - \rho_{k+1,i}^{(1)}$ since it connects with the number of transmitted packets in the first $(i-1)$ transmissions. Thus, we let $\rho_{k,i}^{(2)} + 2T_g + T_{\text{ACK}}$ approximate $\rho_{k+1,i}^{(1)}$,

$$T_i \approx \rho_{k,i+1}^{(1)} - \rho_{k,i}^{(2)} - 2T_g - T_{\text{ACK}} = 2T_d \tag{37}$$

In the same way, for the $M$-th transmission, we also set the penalty term, $\pi(q_M^{(k)})$ and $\pi(q_M^{(k+1)})$, where $\pi(q_M^{(k)})$ is given in (14).

The objective function is also given by

$$J' = \min_{N_i^{(k)}, N_i^{(k+1)}} \sum_{i=1}^{M} \gamma^i T_i \tag{38}$$

This problem is different from the problem in (15). There are two parameters that need to be optimized for each transmission. At the begin of the $i$-th transmission for the $k$-th block, the sender does not receive the feedback message of the $(i-1)$-th transmission for the $(k+1)$-th block. Namely, the sender only knows part of the transmission result of the $(i-1)$-th transmission. Apparently, The standard DP approach is not suitable for this case. We propose a two-step DP approach to solve this problem.

## 6. Proposed Two-Step DP Approach

In each transmission, we separately define the system state for the $k$-th block and the $(k+1)$-th block. For the $k$-th block, the system state is $c_i^{(k)} = (q_{i-1}^{(k)}, h_{i-1}^{(k)}, q_{i-2}^{(k+1)}, h_{i-2}^{(k+1)},$

$B_i^{(k)}$), where $B_i^{(k)}$ is the maximum number of transmitted packets for the $k$-th block. For the $i$-th transmission of the $(k+1)$-th block, the sender already knows the transmission result of the $(i-1)$-th transmission. Thus, the system state is $c_i^{(k+1)} = (q_{i-1}^{(k)}, h_{i-1}^{(k)}, q_{i-1}^{(k+1)}, h_{i-1}^{(k+1)}, B_i^{(k+1)})$.

As analyzed above, the problem for the RLPC-J-ARQ scheme is also a sequential decision-making problem. Thus, it could also be solved based on the DP approach. The main difference is that the $B_i^{(k)}$ and $B_i^{(k+1)}$ depend on the $N_{i-1}^{(k+1)}$ and $N_i^{(k)}$, as given in (32) and (33), respectively. In other words, the state of the $i$-th transmission is not only related to the $(i-1)$-th transmission, but also to $N_i^{(k)}$.

Based on the idea of DP, we propose a two-step DP approach which computes the cost in two steps for each transmission. The proposed method still adopts recursive computing based on the Bellman equation. It is similar to the RLPC-ARQ scheme; we first calculate the cost of the $M$-th transmission, the $i$-th transmission and finally the first transmission. In each transmission, we first calculate the cost of the $(k+1)$-th block. Then, the overall cost of the current transmission is calculated.

*6.1. The Cost of the M-th Transmission*

At the $M$-th transmissions of the $(k+1)$-th block, the system state $c_M^{(k+1)} = (q_{M-1}^{(k)}, h_{M-1}^{(k)}, q_{M-1}^{(k+1)}, h_{M-1}^{(k+1)}, B_M^{(k+1)})$. The expected cost of the $(k+1)$-th block at the $M$-th transmission is similar to (19), which can be computed as

$$J'_M(c_M^{(k+1)}) = \min_{0 \le N_M^{(k+1)} \le B_M^{(k+1)}} \left\{ \gamma^M T_M^{(k+1)} + \mathbb{E}\left[\pi(q_M^{(k+1)}) | c_M^{(k+1)}\right] \right\} \tag{39}$$

where

$$T_M^{(k+1)} = \begin{cases} 0, & \text{if } q_{M-1}^{(k+1)} \ge W \\ T_p N_M^{k+1} + T_\tau, & \text{if } q_{M-1}^{(k+1)} < W \end{cases} \tag{40}$$

is the transmission time duration for the $(k+1)$-th block. The expectation of penalty is given by

$$\mathbb{E}\left[\pi(q_M^{(k+1)}) | c_M^{(k+1)}\right] = \mathbb{E}\left[\pi(q_M^{(k+1)}) | (q_{M-1}^{(k+1)}, h_{M-1}^{(k+1)})\right]$$

$$= C \sum_{s_M^{(k+1)}=0}^{u_M^{(k+1)}} \sum_{l=1}^{L} P_s(s_M^{(k+1)}, h_M^{(k+1)}, N_M^{(k+1)}) P(h_M^{(k+1)} = \delta_l | h_{M-1}^{(k+1)}) \tag{41}$$

Next, we consider the expected cost for the overall $M$-th transmission, $J'_M(c_M^{(k)})$. At this point, the sender knows the system state $c_M^{(k)} = (q_{M-1}^{(k)}, h_{M-1}^{(k)}, q_{M-2}^{(k+1)}, h_{M-2}^{(k+1)}, B_M^{(k)})$.

It is clear that the calculation of $J'_M(c_M^{(k)})$ highly depends on $q_{M-1}^{(k)}$ and $q_{M-2}^{(k+1)}$. The state is different if $q_{M-1}^{(k)}$ and $q_{M-2}^{(k+1)}$ are larger or less than $M$. Thus, the calculation of $J'_M(c_M^{(k)})$ is different with different $q_{M-1}^{(k)}$ and $q_{M-2}^{(k+1)}$. So, the analysis of different situations is as follows.

If $q_{M-1}^{(k)} \ge W$ and $q_{M-2}^{(k+1)} \ge W$, it is obvious that

$$J'_M(c_M^{(k)}) = 0 \tag{42}$$

If $q_{M-1}^{(k)} \ge W$ and $q_{M-2}^{(k+1)} < W$, the transmission of the $k$-th block is finished. Then, $N_M^{(k)} = 0$. It degenerates into the problem in the RLPC-ARQ scheme. At this time, only the $(k+1)$-th block may need to be transmitted. It relies on the transmission result of the $(M-1)$-th transmission for the $(k+1)$-th block. According to (32), $N_{M-1}^{(k+1)} = B_M^{(k)}$. Thus,

$$J'_M(c_M^{(k)}) = \min_{0 \leq N_M^{(k+1)} \leq Q} \mathbb{E}[J_M(h_{M-1}^{(k+1)}, q_{M-1}^{(k+1)}) | (h_{M-2}^{(k+1)}, q_{M-2}^{(k+1)}, B_M^{(k)})]$$

$$= \sum_{s_{M-1}^{(k+1)}=0}^{B_M^{(k)}} \sum_{l=0}^{L} J_M(h_{M-1}^{(k+1)}, q_{M-1}^{(k+1)}) P_s(s_{M-1}^{(k+1)}, h_{M-1}^{(k+1)}, B_M^{(k)}) P(h_{M-1}^{(k+1)} = \delta_l | h_{M-2}^{(k+1)}) \quad (43)$$

Note that $q_{M-1}^{(k+1)} = q_{M-2}^{(k+1)} + s_{M-1}^{(k+1)}$.

If $q_{M-1}^{(k)} < W$ and $q_{M-2}^{(k+1)} \geq W$, then $B_M^{(k)} = Q$. This problem also degenerates into the former problem.

$$J'_M(c_M^{(k)}) = \min_{0 \leq N_M^{(k+1)} \leq Q} \left\{ \mathbb{E}[T_M^{(k)} | c_M^{(k)}] + \mathbb{E}\left[\pi(q_M^{(k)})\right] \right\}$$

$$= J_M(h_{M-1}^{(k)}, q_{M-1}^{(k)}) \quad (44)$$

If $q_{M-1}^{(k)} < W$ and $q_{M-2}^{(k+1)} < W$, then the expected cost for the overall $M$-th transmission is given by

$$J'_M(c_M^{(k)}) = \min_{0 \leq N_M^{(k)} \leq B_M^{(k)}} \left\{ \gamma^M \mathbb{E}[T_M^{(k)} | c_M^{(k)}] + \mathbb{E}\left[\pi(q_M^{(k)}) | c_M^{(k)}\right] + \mathbb{E}[J'_M(c_M^{(k+1)}) | c_M^{(k)}] \right\} \quad (45)$$

The first and second term in (45) are the cost from the $k$-th block; the third term is from the $(k+1)$-th block.

As analyzed in (35), if $q_{M-1}^{(k+1)} < W$, the number of $N_M^{(k)}$ does not affect the cost. Thus, in this case,

$$\mathbb{E}[T_M^{(k)} | c_M^{(k)}] = 0 \quad (46)$$

If $q_{M-1}^{(k+1)} \geq W$, then $N_M^{(k+1)} = 0$. According to (37), the expected additional transmission time duration for the $M$-th transmission of the $k$-th block is $2T_d$. The corresponding probability is given by

$$P(q_{M-1}^{(k+1)} \geq W) = \sum_{s_M^{(k+1)}=u_{M-1}^{(k+1)}}^{B_M^{(k)}} P_s(s_{M-1}^{(k+1)}, h_{M-1}^{(k+1)}, B_M^{(k)}) \quad (47)$$

To summarize, if $q_{M-1}^{(k)} < W$ and $q_{M-2}^{(k+1)} < W$, the first term of (45) is

$$\mathbb{E}[T_M^{(k)} | c_M^{(k)}] = \mathbb{E}[T_M^{(k)} | (q_{M-2}^{(k+1)}, h_{M-2}^{(k+1)}, B_M^{(k)})]$$

$$= 2T_d \sum_{s_M^{(k+1)}=u_{M-1}^{(k+1)}}^{B_M^{(k)}} \sum_{l=1}^{L} P_s(s_{M-1}^{(k+1)}, h_{M-1}^{(k+1)}, B_M^{(k)}) P(h_{M-1}^{(k+1)} = \delta_l | h_{M-2}^{(k+1)}) \quad (48)$$

The second term of (45) is similar to (41). The third term of (45) is

$$\mathbb{E}[J'_M(c_M^{(k+1)}) | c_M^{(k)}] = \mathbb{E}[J'_M(c_M^{(k+1)}) | (q_{M-2}^{(k+1)}, h_{M-2}^{(k+1)}, B_M^{(k)})]$$

$$= \sum_{s_{M-1}^{(k+1)}=0}^{B_M^{(k)}} \sum_{l=0}^{L} J'_M(c_M^{(k+1)}) P_s(s_{M-1}^{(k+1)}, h_{M-1}^{(k+1)}, B_M^{(k)}) P(h_{M-1}^{(k+1)} = \delta_l | h_{M-2}^{(k+1)}) \quad (49)$$

In summary, the cost of the $M$-th transmission is given as

$$J'_M(c_M^{(k)}) = \begin{cases} (42), & \text{if } q_{i-1}^{(k)} \geq W, q_{M-2}^{(k+1)} \geq W \\ (43), & \text{if } q_{M-1}^{(k)} \geq W, q_{M-2}^{(k+1)} < W \\ (44), & \text{if } q_{M-1}^{(k)} < W, q_{M-2}^{(k+1)} \geq W \\ (45), & \text{if } q_{M-1}^{(k)} < W, q_{M-2}^{(k+1)} < W \end{cases} \tag{50}$$

### 6.2. The Cost of the i-th Transmission

For the $i$-th transmission, the system state for the $(k+1)$-th block is $c_i^{(k+1)} = (q_{i-1}^{(k)}, h_{i-1}^{(k)}, q_{i-1}^{(k+1)}, h_{i-1}^{(k)}, B_i^{(k+1)})$. The expected cost incurred from the $i$-th transmission of the $(k+1)$-th block to termination is given as Equation (51).

$$J'_i(c_i^{(k+1)}) = \begin{cases} 0, & \text{if } q_{i-1}^{(k)} \geq W, q_{i-1}^{(k+1)} \geq W \\ J_i(h_{i-1}^{(k+1)}, q_{i-1}^{(k+1)}), & \text{if } q_{i-1}^{(k)} \geq W, q_{i-1}^{(k+1)} < W \\ \mathbb{E}\left[ J_{i+1}(h_i^{(k)}, q_i^{(k)}) | (h_{i-1}^{(k)}, q_{i-1}^{(k)}) \right], & \text{if } q_{i-1}^{(k)} < W, q_{i-1}^{(k+1)} \geq W \\ \min_{N_i^{(k+1)}} \left\{ \gamma^i(T_i^{(k+1)}) + \gamma \mathbb{E}\left[ J'_{i+1}(c_{i+1}^{(k)}) | c_i^{(k+1)} \right] \right\} & \text{if } q_{i-1}^{(k)} < W, q_{i-1}^{(k+1)} < W \end{cases} \tag{51}$$

The conditional expectation in (51) is given as

$$\mathbb{E}\left[ J'_{i+1}(c_{i+1}^{(k)}) | c_i^{(k+1)} \right] = \sum_{s_i^{(k)}=0}^{N_i^{(k)}} \sum_{l=0}^{L} J'_{i+1}(c_{i+1}^{(k)}) P_s(s_i^{(k)}, h_i^{(k)}, N_i^{(k)}) P(h_i^{(k)} = \delta_l | h_{i-1}^{(k)}) \tag{52}$$

It is interesting that (52) does not seem to have much to do with $N_i^{(k+1)}$. The transition probability mainly depends on the channel state $h_{i-1}^{(k)}$ and the transmitted packets of $N_i^{(k)}$. However, $N_i^{(k+1)}$ affects $J'_{i+1}(c_{i+1}^{(k)})$ by limiting the range of $N_{i+1}^{(k)}$ in (52).

For the $k$-th block, the system state is $c_i^{(k)} = (q_{i-1}^{(k)}, h_{i-1}^{(k)}, q_{i-2}^{(k+1)}, h_{i-2}^{(k)}, B_i^{(k)})$. The expected cost incurred from the $i$-th transmission to termination is given by (53). Similarly, we have

$$J'_i(c_i^{(k)}) = \begin{cases} 0, & \text{if } q_{i-1}^{(k)} \geq W, q_{i-2}^{(k+1)} \geq W \\ J_i(h_{i-1}^{(k)}, q_{i-1}^{(k)}), & \text{if } q_{i-1}^{(k)} < W, q_{i-2}^{(k+1)} \geq W \\ \mathbb{E}\left[ J_i(h_{i-1}^{(k+1)}, q_{i-1}^{(k+1)}) | (h_{i-2}^{(k+1)}, q_{i-2}^{(k+1)}) \right], & \text{if } q_{i-1}^{(k)} \geq W, q_{i-2}^{(k+1)} < W \\ \min_{N_i^{(k)}} \left\{ \gamma^i \mathbb{E}\left[ T_i^{(k)} | c_i^{(k+1)} \right] + \mathbb{E}\left[ J'_i(c_i^{(k)}) | c_i^{(k+1)} \right] \right\} & \text{if } q_{i-1}^{(k)} < W, q_{i-2}^{(k+1)} < W \end{cases} \tag{53}$$

$$\mathbb{E}\left[ J'_i(c_i^{(k+1)}) | c_i^{(k)} \right] = \sum_{s_{i-1}^{(k+1)}=0}^{N_{i-1}^{(k+1)}} \sum_{l=0}^{L} J'_i(c_i^{(k+1)}) P_s(s_{i-1}^{(k+1)}, h_{i-1}^{(k+1)}, N_{i-1}^{(k+1)}) P(h_{i-1}^{(k+1)} = \delta_l | h_{i-2}^{(k+1)}) \tag{53}$$

$N_i^{(k)}$ also has no influence on transfer probability and affects $J'_i(c_i^{(k+1)})$ by limiting the range of $N_i^{(k+1)}$.

*6.3. The Cost of the First Transmission*

For $i = 1$, $\boldsymbol{c}_1^{(k+1)} = (q_0^{(k)}, h_0^{(k)}, q_0^{(k+1)}, h_0^{(k+1)}, B_1^{(k+1)})$. Note that $q_0^{(k)} = q_0^{(k+1)} = 0$ and the transmitter have no feedback information about the channels. Thus, we use the steady-state probability to present the probability. Then, we have

$$J_1'(\boldsymbol{c}_1^{(k+1)}) = \min_{N_1^{(k+1)}} \left\{ \gamma(T_1^{(k+1)}) + \mathbb{E}\left[ J_2'(\boldsymbol{c}_2^{(k)}) | \boldsymbol{c}_1^{(k+1)} \right] \right\} \tag{54}$$

where

$$\mathbb{E}\left[ J_2'(\boldsymbol{c}_2^{(k)}) | \boldsymbol{c}_1^{(k+1)} \right] = \sum_{s_1^{(k)}=0}^{N_1^{(k)}} \sum_{l=0}^{L} J_2'(\boldsymbol{c}_2^{(k)}) P_s(s_1^{(k)}, h_1^{(k)}, N_1^{(k)}) P_\pi(h_1^{(k)} = \delta_l) \tag{55}$$

Similarly, $\boldsymbol{c}_1^{(k)} = (q_0^{(k)}, h_0^{(k)}, q_{-1}^{(k+1)}, h_{-1}^{(k)}, B_1^{(k)})$; the cost for the overall $M$ transmissions is given as

$$\begin{aligned} J_1'(\boldsymbol{c}_1^{(k)}) &= \min_{N_m^{(k)}} \left\{ \gamma(T_1^{(k)}) + \mathbb{E}\left[ J_1'(\boldsymbol{c}_1^{(k+1)}) | \boldsymbol{c}_1^{(k)} \right] \right\} \\ &= \min_{N_m^{(k)}} \left\{ \gamma(T_p N_1^{(k+1)} + 2T_g + T_{\text{ACK}}) + \mathbb{E}\left[ J_1'(\boldsymbol{c}_1^{(k+1)}) | \boldsymbol{c}_1^{(k)} \right] \right\} \end{aligned} \tag{56}$$

where

$$\mathbb{E}\left[ J_1'(\boldsymbol{c}_1^{(k+1)}) | \boldsymbol{c}_1^{(k)} \right] = \sum_{l=0}^{L} J_2'(0, h_1^{(k)}, 0, h_0^{(k+1)}, N_1^{(k)}) P_\pi(h_1^{(k+1)} = \delta_l) \tag{57}$$

The two-step DP approach also first finds the optimal solution for each state and records it into a table. Then, the sender chooses the optimal $N_i^{(k)}$ and $N_i^{(k+1)}$ according to system state. The details of the proposed optimal two-step DP policy are summarized in Algorithm 1.

---

**Algorithm 1** The proposed two-step DP algorithm.

---

1: Let $\boldsymbol{c}_M^{(k+1)} = (q_{M-1}^{(k)}, h_{M-1}^{(k)}, q_{M-1}^{(k+1)}, h_{M-1}^{(k)}, B_M^{(k+1)})$.
2: Calculate $J'_M(\boldsymbol{c}_M^{(k+1)})$, $\forall q_{M-1}^{(k)}$, $\forall h_{M-1}^{(k)}$, $\forall q_{M-1}^{(k+1)}$, $\forall h_{M-1}^{(k+1)}$, $\forall B_M^{(k+1)}$, based on (39) and save in the table.
3: Let $\boldsymbol{c}_M^{(k)} = (q_{M-1}^{(k)}, h_{M-1}^{(k)}, q_{M-2}^{(k+1)}, h_{M-2}^{(k)}, B_M^{(k)})$.
4: Calculate $J'_M(\boldsymbol{c}_M^{(k)})$, $\forall q_{M-1}^{(k)}$, $\forall h_{M-1}^{(k)}$, $\forall q_{M-2}^{(k+1)}$, $\forall h_{M-2}^{(k+1)}$, $\forall B_M^{(k)}$, based on (50) and save in the table.
5: $i = M - 1$.
6: **while** $i > 0$
7:    Let $\boldsymbol{c}_i^{(k+1)} = (q_{i-1}^{(k)}, h_{i-1}^{(k)}, q_{i-1}^{(k+1)}, h_{i-1}^{(k)}, B_i^{(k+1)})$.
8:    Calculate $J'_i(\boldsymbol{c}_i^{(k+1)})$, $\forall q_{i-1}^{(k)}$, $\forall h_{i-1}^{(k)}$, $\forall q_{i-1}^{(k+1)}$, $\forall h_{i-1}^{(k)}$, $\forall B_i^{(k+1)}$ based on (51) and save in the table
9:    Let $\boldsymbol{c}_i^{(k)} = (q_{i-1}^{(k)}, h_{i-1}^{(k)}, q_{i-2}^{(k+1)}, h_{i-2}^{(k)}, B_i^{(k)})$.
10:    Calculate $J'_i(\boldsymbol{c}_i^{(k)})$, $\forall q_{i-1}^{(k)}$, $\forall h_{i-1}^{(k)}$, $\forall q_{i-2}^{(k+1)}$, $\forall h_{i-2}^{(k)}$, $\forall B_i^{(k)}$ based on (53) and save in the table
11:    Find the optimal $(N_i^{(k)}, N_i^{(k+1)})$ to maximize $J'_i(\boldsymbol{c}_i^{(k+1)})$
12:    $i = i - 1$;
13: **end while**
14: **for** $i = 1; i \le M; i = i + 1$ **do**
15:    Find the optimal $(N_i^{(k)}, N_i^{(k+1)})$ based on line 9.
16:    Update the $\boldsymbol{c}_i^{(k)}$ and $\boldsymbol{c}_i^{(k+1)}$.
17: **end for**

---

## 7. Simulation Results and Discussions

In this section, we analyze the performance of the proposed packet-coding scheme with the SR ARQ scheme. In the SR ARQ scheme, the transmitter sends $W$ packets in each transmission and waits for the ACK messages. The transmitter retransmits the failed packets and the new packets to form a new group of packets in the next transmission.

Unless otherwise specified, the simulation parameters of the system are presented in Table 1. We consider BPSK modulation and the UWA channels as a Markov Chain model. The communication link often exhibits ON/OFF behaviour in UWA communications. Therefore, we could model the channels as a two-state Markov Chain model [22,24]: "Bad (0)" and "Good (1)". The BERs under the good and the bad channel conditions are $e_0 = 1 \times 10^{-3}$ and $e_1 = 5 \times 10^{-4}$, respectively. The transition matrix of the channel state is [24]

$$P_h = \begin{bmatrix} P_{00} & P_{01} \\ P_{10} & P_{11} \end{bmatrix} \tag{58}$$

The steady-state probabilities are given by $\pi_0 = P_{10}/(P_{10} + P_{01})$, whereas $\pi_1 = 1 - \pi_0$. The steady-state probabilities could be used as the probabilities of CSI at the initial transmission, which does not have the feedback messages. In addition, we define a memory parameter $\mu = 1 - (P_{10} + P_{01})$ to present the time correlation of channel states. Then, the transition matrix of the channel state can be computed by $\pi_0$ and $\mu$. In the simulation, we set $\pi_0 = 0.43$ and $\mu = 0.79$ according to the experiment data given in [23].

The PEP $P_c$ depends on the length of packet $N_b$ and the BER $e$; it can be calculated from the equilibrium distribution of the generalized Pareto renewal process [5]

$$P_c = \left( \frac{1 - \theta}{1 - \theta + eN_b\theta} \right)^{-1 + \frac{1}{\theta}} \tag{59}$$

where $\theta$ is the shape parameter of the generalized Pareto distribution. In this paper, we use $\theta = 0.12$.

For the J-ARQ scheme, the number of transmitted groups during a round-trip time is given by

$$N_g = \frac{2T_d}{WT_p + 2T_g + T_{ACK}} \tag{60}$$

**Table 1.** Simulation parameters.

| Parameter | Value |
|:---:|:---:|
| Symbol duration $T_s$ | 0.2 ms |
| Transmission distance $d$ | 5 km |
| The length of a packet $N_b$ | 1000 |
| Guard interval $T_g$ | $T_sN_b$ |
| Time duration of ACK $T_{ACK}$ | $T_sN_b$ |
| Discount factor $\gamma$ | 1.01 |
| The max number of transmissions $M$ | 5 |
| Penalty term $C$ | 1000 |
| Block size $W$ | 10 |

We evaluate the performance based on the throughput efficiency, which is given by

$$\eta = (1 - P_E)\frac{N_bWT_s}{T_{\text{total}}} \tag{61}$$

where $P_E$ is the probability of packet loss; $T_{\text{total}}$ is the total transmission time duration.

As shown in [5], the performance of the J-ARQ scheme depends on the number of source data. Thus, we first study the impact of the number of source data. Figure 3 illustrates the throughput efficiency as a function of the number of source data packets $W_s$. From Figure 3, we find the SR-ARQ scheme has the lowest throughput efficiency in all schemes. The throughput efficiency performance of conventional J-ARQ increases with the length of source data. If the sender has enough source data, the J-ARQ scheme has the best throughput efficiency. The performance of the RLPC-ARQ scheme and RLPC-J-ARQ is independent of data length. The throughput efficiency of RLPC-J-ARQ is better than RLPC-ARQ since it takes full advantage of the transmission time. When the data length is short, the performance of J-ARQ is worse than that of the RLPC-J-ARQ scheme. In the RLPC-J-ARQ scheme, the transmission duration is adjustable. Thus, it has a better performance when $W_s$ is not big enough.

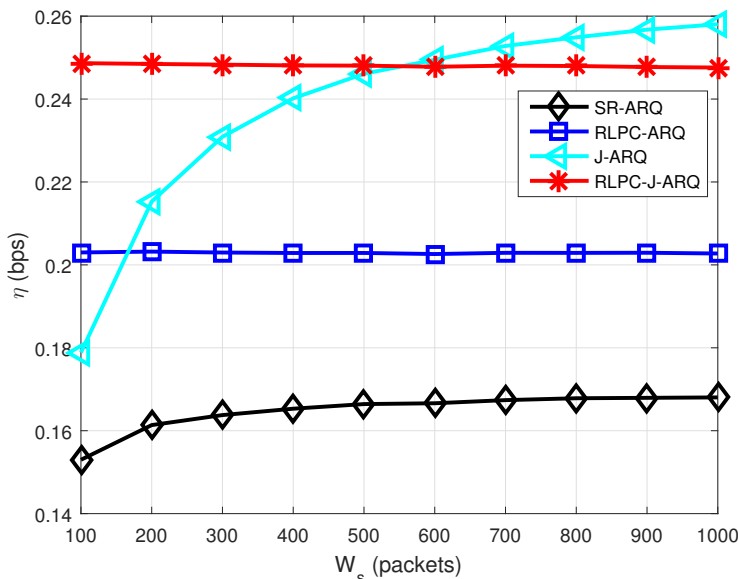

**Figure 3.** Throughput efficiency as a function of the number of source data $W_s$.

Figure 4 illustrates the throughput efficiency as a function of block size $W$. In this simulation, $W_s = 400$. The throughput efficiency of all schemes increases with the increase of $W$. The proposed RLPC schemes are still better than the corresponding ARQ schemes. That makes sense since a bigger $W$ means the transmitter could send more packets during a transmission; it reduces the number of ACKs. There is a turning point for the J-ARQ scheme. This is because $N_g$ will decrease with the increase in group size $W$. When $W = 10$, $N_g = 4$ and it turns to $N_g = 3$ for $W = 12$. The throughput efficiency of the J-ARQ scheme decreases as $N_g$ becomes smaller since it leads to more idle time.

Figure 5 shows the impact of transmission distance on throughput efficiency. Naturally, the throughput efficiency decreases with the increase of $d$, since it leads to an increase in propagation delay $T_d$. The performance of the RLPC-ARQ scheme is always better than the SR-ARQ scheme. For the J-ARQ scheme, the throughput efficiency becomes larger at $d = 5000$ m because $N_g$ becomes larger. The transmitter sends two group packets during a round-trip time with $d = 4500$ m, while it sends three groups with $d = 5000$ m. The same situation also occurs at $d = 7000$ m. For the RLPC-J-ARQ scheme, the performance continuously decreases since we only consider that two blocks are transmitted simultaneously. However, its throughput efficiency is still better than that of the J-ARQ scheme when $d$ is less than 7000 m.

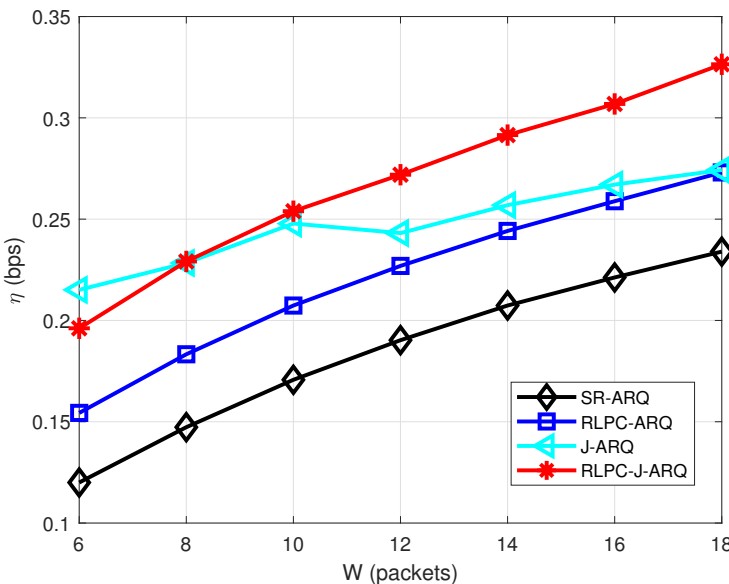

**Figure 4.** Throughput efficiency as a function of the number of group packets *W*.

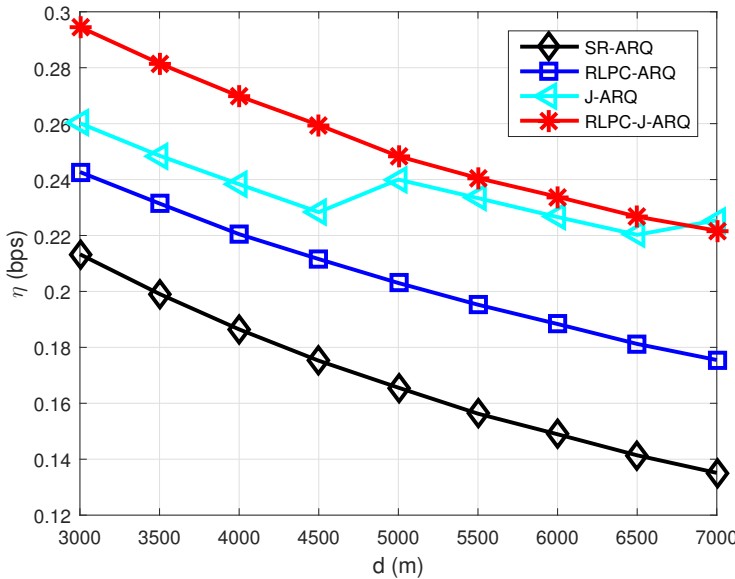

**Figure 5.** Throughput efficiency as a function of transmission distance *d*.

Finally, we analyze the impact of steady-state probability $\pi_0$, which reflects the effect of UWA channels. Figure 6 illustrates the throughput efficiency as a function of steady-state probability $\pi_0$. From Figure 6, we find the performance of all schemes will deteriorate with the increase of $\pi_0$. A larger $\pi_0$ means the channel is more likely to be bad. The performance gain between with RLPC and without RLPC also increase as $\pi_0$ increase. This is because the RLPC schemes under the bad channel could improve the successful decode probability by increasing the transmitted packets to reduce the retransmission. However, the conventional ARQ schemes need to retransmit many times. Thus, the superiority of RLPC schemes is more reflected in the case of poor channels. After all, the conventional ARQ scheme also does not need to be retransmitted when the channel is good.

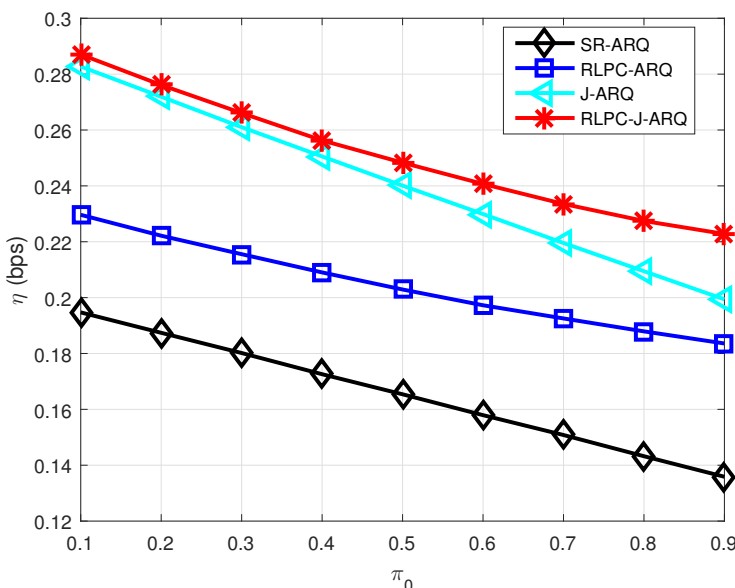

**Figure 6.** Throughput efficiency as a function of the steady-state probability $\pi_0$.

This paper proposed two ARQ schemes with RLPC for UWA communications. The limitations of the proposed method are as follows. First, the sender knows the transition matrix of the channel state in advance, which needs to do a lot of experiments. Second, the proposed methods were based on the DP algorithm. The computational complexity is very high with a large system state. Thus, it is better to solve the problem with deep learning method.

## 8. Conclusions

This work investigates the adaptive packet coding for reliable UWA communications. This paper proposes two schemes based on two different ARQs. The first scheme is based on the conventional S&W ARQ scheme. The sender chooses the optimal number of packets in each transmission according to the feedback transmission result about the last transmission. We also set maximum retransmission times to avoid infinite retransmission considering the error-prone characteristic of UWA channels. This problem is formulated as a finite horizon optimization and solved with the DP algorithm. To overcome the impact of long propagation delay, we propose the modified juggling-like ARQ scheme for RLPC. Compared with the standard J-ARQ scheme, the transmission duration of the proposed scheme is variant and adapts to the rateless characteristics of the random linear packet coding. A two-step DP algorithm is proposed to find out the optimal solution. Simulation results demonstrate the performance gain of the proposed schemes as well as the impact of various practical factors such as data length, group size, channel steady-state distribution and transmission distance.

**Author Contributions:** Conceptualization, L.J. and Y.T.; formal analysis, L.J.; methodology, L.J. and H.Y.; software, L.J. and Y.T.; validation, C.H. and H.Y.; writing—original draft, L.J.; writing— review and editing, H.Y. and C.H.; visualization, L.J. and Y.T.; supervision, H.Y.; project administration, L.J. and H.Y.; funding acquisition, L.J. All authors have read and agreed to the published version of the manuscript.

**Funding:** This research was funded by the National Natural Science Foundation of China (61801079, 62071383, 61871418), the open research fund of State Key Laboratory of Integrated Services Networks (ISN22-15), and the Science and Technology on Underwater Information and Control Laboratory (6142218200408).

**Institutional Review Board Statement:** Not applicable.

**Informed Consent Statement:** Not applicable.

**Data Availability Statement:** The data presented in this paper are available after contacting the corresponding author.

**Acknowledgments:** The authors would like to thank the anonymous reviewers for their careful reading and valuable comments.

**Conflicts of Interest:** The authors declare no conflict of interest.

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
