# Peer review of "Adaptive Packet Coding for Reliable Underwater Acoustic Communications"

_remotesensing, doi:10.3390/rs14194712_

Round 1

Reviewer 1 Report

The paper proposes an adaptive rateless coding scheme to be coupled with a scheme named juggling-like ARQ, which leverages the long propagation delay of underwater communication channels in order to interleave data and ACK packets, and thus improve the efficiency of ARQ schemes beyond those of stop-and-wait ARQ. In the scheme proposed by the authors, the data packets exchanged are rateless-coded packets. A two-step DP algorithm is proposed to find out the optimal solutions for this case. Simulation results show that the proposed schemes can improve the throughput efficiency and reduce the outage probability.

The paper has a good logical flow and readability. The overall content is complete and innovative.

This paper can be considered for publication after some revisions.

1 The difference between the proposed RLPC-J-ARQ scheme and the J-ARQ is not given clearly.

2 In Fig.3, the performance of J-ARQ is better than the proposed RLPC-J-ARQ. Could you explain the reason?

3 There are some spelling and grammar mistakes in this paper. 

Author Response

1 The difference between the proposed RLPC-J-ARQ scheme and the J-ARQ is not given clearly.

Response: Thanks for your comment. In J-ARQ, it just uses the rateless code to generate some packets to avoid wasting the available transmission time when there is not sufficient data at the transmitter. However, it does not consider the optimal number of transmission packets. And the rateless code is only used when there is not enough data to transmit to avoid idle. In addition, it only considers the time-invariant channels.

In our scheme, the transmission duration is not fixed, it is adjustable based on the channel state and the transmission result. 

2 In Fig.3, the performance of J-ARQ is better than the proposed RLPC-J-ARQ. Could you explain the reason?

Response: Thanks for your comment. The main drawback of J-ARQ scheme is that the J-ARQ scheme has a fixed transmission duration. Then, if there are not enough data at one transmission, it has to transmit the rateless code packets to avoid idle. It decreases the transmission efficiency.  If there are enough transmission data at the sender, the effect of the long transmission propagation time is not obvious. In each transmission, the sender always has the data to transmit, and there is no idle state. For our proposed scheme, we adopt an adjustable transmission duration by exploiting the characteristics of the rateless code.

3 There are some spelling and grammar mistakes in this paper.

Response: Thanks for your careful review. We have modified the grammar and typos carefully in the revised manuscript.

Reviewer 2 Report

The paper deals with the problem of adaptive packet coding for reliable Under Water Acoustic (UWA) communications.

The numerical results show the advantage provided by the proposed scheme, in particular as a function of the distance and steady-state probability of the channel model.

The main points to be addressed:

- In Sect. 3 it is presented the formulation of the problem and the procedure for achieving the optimal solution. However, it is should be clarified further how this optimal solution and the procedure are related to the proposed approaches in Sect. 4 and 5.

- The readibility of Sect. 4 and 5 could be enhanced also by including some flowcharts of the processes.

- A revision of the text is also needed for correcting some typos.

Author Response

- In Sect. 3 it is presented the formulation of the problem and the procedure for achieving the optimal solution. However, it is should be clarified further how this optimal solution and the procedure are related to the proposed approaches in Sect. 4 and 5.

Response: Thanks for your comment. The problem for RLPC-J-ARQ scheme is also a sequential sequential decision making problem. Thus, it can be solved by recursive computing based on the Bellman equation. The main difference between the problem of RLPC-J-ARQ scheme and the RLPC-ARQ scheme is that the number of transmitted packets in each transmission is monotone decreasing in RLPC-J-ARQ scheme and there is no constrain about the number of transmitted packets in RLPC-ARQ scheme. In other words, the state of i-th transmission is not only related to the (i-1)-th transmission but also to the N_{i}^{(k)}. They are the same expect this difference. We have added the explanation in the revised manuscript.  

- The readibility of Sect. 4 and 5 could be enhanced also by including some flowcharts of the processes.

Response: Thanks for your comment. It seems complicated only because the state of c_i^k and c_i^{k+1} is different with the value of q_{i-1}^{(k)} and q_{i-1}^{(k+1)}. To make it clear, we have added an algorithm table to summarize the proposed algorithm as you suggested. 

- A revision of the text is also needed for correcting some typos

Response: Thanks for your careful review. We have modified the grammar and typos carefully in the revised manuscript.

Reviewer 3 Report

1) v.16 "0. Introduction"- probably the numbers should start from <<1>>
2) v.108 "1. system model" - one should use capital letters
3) v.114 "which the receiver" - probably one omits words <<means>> i.e. "which means the receiver"
4) eq.(1) - $N_b$ is not clear; Is this a number of bits or symbols? Probably symbols because $T_s$ is a symbol duration. Also, it is not clear if $N_b$ is a constant number?
v.20 - "if there are enough coded packets are successfully received." - one of the <<are>> should be removed
5) v.121-123 - The sentence is confusing. I'm not familiar with ARQ and RLPC, but despite this, for the readers, it would be hard to catch what does it mean "new code packets". Is this a part of the information? or a code/key needed for coding/encoding? Consequently - what does it mean "the same original packets"?
6) v.161 - IMO - not each transmission time increases, because part of the packed was received and was decoded in the previous transmission. So, the best solution is to send as many coded packed as possible at the first transmission.
7) v.189-190 "a guard interval is reserved for the node to transfer the mode" please consider "to switch" or "to change" the mode
8) v.202 "This problem is different from the above one" - please consider pointing out which exact problem you have in mind; probably eq.(15)
9) v.271 "It is naturally the throughput efficiency decreases with the increase of d." - have you got any mathematical model of such decreasing? Eq.(61) does not point out any influence of distance on throughput efficiency.
10) v.277 "its throughput efficiency still better than" probably you omit the verb <<is>>
11) Fig.3 - Is (bps) mean bit-per-second? IMO eq.(61) gives the unit bit or symbols depending on the meaning of $N_b$

Author Response

1)16 "0. Introduction"- probably the numbers should start from <<1>>

2) v.108 "1. system model" - one should use capital letters

3) v.114 "which the receiver" - probably one omits words <<means>> i.e. "which means the receiver"

4) eq.(1) - $N_b$ is not clear; Is this a number of bits or symbols? Probably symbols because $T_s$ is a symbol duration. Also, it is not clear if $N_b$ is a constant number?

v.20 - "if there are enough coded packets are successfully received." - one of the <<are>> should be removed

5) v.121-123 - The sentence is confusing. I'm not familiar with ARQ and RLPC, but despite this, for the readers, it would be hard to catch what does it mean "new code packets". Is this a part of the information? or a code/key needed for coding/encoding? Consequently - what does it mean "the same original packets"?

6) v.161 - IMO - not each transmission time increases, because part of the packed was received and was decoded in the previous transmission. So, the best solution is to send as many coded packed as possible at the first transmission.

7) v.189-190 "a guard interval is reserved for the node to transfer the mode" please consider "to switch" or "to change" the mode

8) v.202 "This problem is different from the above one" - please consider pointing out which exact problem you have in mind; probably eq.(15)

9) v.271 "It is naturally the throughput efficiency decreases with the increase of d." - have you got any mathematical model of such decreasing? Eq.(61) does not point out any influence of distance on throughput efficiency.

10) v.277 "its throughput efficiency still better than" probably you omit the verb <<is>>

11) Fig.3 - Is (bps) mean bit-per-second? IMO eq.(61) gives the unit bit or symbols depending on the meaning of $N_b$

Response:

Thanks for your careful review. We have modified the grammar and typos carefully in the revised manuscript.

 As for comment 5, our response is as follows. According to the characteristic of the RLPC, it could generate infinite coded packets based on the original information packets. The receiver could recover the information packets if it successfully receives enough coded packets. The new coded packets means the coded packets that are also generated from the information packets. In our scheme, the sender would not retransmit the unsuccessful packets. It sends other packets which are also generated from the information packets. We have modified the sentence to make it clear in the revised manuscript.

As for comment 6, our response is as follows. You are right that we should send as many coded packs as possible at the first transmission. However, considering the characteristic of underwater acoustic channels, there still exists the risk that the receiver receives enough packets successfully. Meanwhile, if we send too many packets in each transmission, the sending time also increases, which also decreases efficiency. We have modified the sentence to make it clear in the revised manuscript.

As for comment 9, our response is as follows. With the increase of d, the propagation delay T_d increases. It also leads to the total transmission time duration T_{total} increases. Thus, the throughput efficiency decreases. We have modified the sentence to make it clear in the revised manuscript.

As for comment 11 and comment 4, our response is as follows. $N_b$ denotes the number of symbols and should be constant during the transmission. In the simulation section, we consider BPSK modulation, which means that the duration of the symbol and bit are the same. We have modified the sentence to make it clear in the revised manuscript.

As for other comments, we have made changes based on your comments. Thank you again for your careful reading of the paper.